# Thermoelectric signature of quantum critical phase in a doped spin-liquid candidate

K. Wakamatsu [1], Y. Suzuki [1], T. Fujii [2], K. Miyagawa [1], H. Taniguchi [3] & K. Kanoda [1,4,5,6] ✉

Quantum spin liquid is a nontrivial magnetic state of longstanding interest, in which spins are strongly correlated and entangled but do not order; further intriguing is its doped version, which possibly hosts strange metal and unconventional superconductivity. A promising candidate of the doped spin liquid is a triangular-lattice organic conductor, $\kappa$-(BEDT-TTF)$_4$Hg$_{2.89}$Br$_8$, recently found to hold metallicity, spin-liquid-like magnetism, and BEC-like superconductivity. The nature of the metallic state with the spin-liquid behaviour is awaiting to be further clarified. Here, we report the thermoelectric signature that mobile holes in the spin liquid background are in a quantum critical state and it pertains to the BEC-like superconductivity. The Seebeck coefficient divided by temperature, $S/T$, is enhanced on cooling with logarithmic divergence indicative of quantum criticality. Furthermore, the logarithmic enhancement is correlated with the superconducting transition temperature under pressure variation, and the temperature and magnetic field profile of $S/T$ upon the superconducting transition change with pressure in a consistent way with the previously suggested BEC-BCS crossover. The present results reveal that the quantum criticality in a doped spin liquid emerges in a phase, not at a point, and is involved in the unconventional BEC-like nature.

Strong correlation among electrons brings about various emergent phenomena in solids. Among them, quantum criticality has long been a focus of profound interest since strange metal, unconventional superconductivity, and magnetic quantum phase transition all spring from a single point called a quantum critical point as extensively discussed in heavy electron systems, copper oxides, and iron pnictides[1–3]. Notably, recent studies of heavy electron compounds have found quantum critical phases residing in a range of parameter space, not at a point, suggesting that a non-Fermi-liquid phase is stabilised[4–6] along with a quantum spin liquid (QSL) of $f$ electrons[7]. The relation between the quantum critical phase and frustration has recently attracted intense attention[8,9]. Thus, a QSL, which is an exotic state of longstanding interest arising from spin frustration[10–12], can offer a novel

stage for quantum critical phenomena if the QSL acquires itinerancy in the charge degrees of freedom by doping[13]. In this connection, it is notable that the organic conductor, $\kappa$-(BEDT-TTF)$_4$Hg$_{2.89}$Br$_8$ (abbreviated as $\kappa$-HgBr) is suggested to be a doped QSL that hosts a non-Fermi-liquid phase in a finite pressure range.

$\kappa$-HgBr is a layered compound consisting of conducting BEDT-TTF layers with a nearly isotropic triangular lattice of BEDT-TTF dimers with the transfer integral ratio, $t'/t$, of 1.02 (Fig. 1a, b) and insulating Hg$_{2.89}$Br$_8$ layers. The nonstoichiometry of Hg comes from an incommensurate lattice against a BEDT-TTF lattice and the missing content from 3.0 contributes 11% hole doping to a half-filled band[14]. Remarkably, $\kappa$-HgBr shows non-Fermi liquidity and spin susceptibility well scaled to that of the spin-liquid

[1]Department of Applied Physics, University of Tokyo; Bunkyo-ku, Tokyo 113-8656, Japan. [2]Cryogenic Research Center, University of Tokyo; Bunkyo-ku, Tokyo 113-0032, Japan. [3]Graduate School of Science and Engineering, Saitama University, Saitama 338-8570, Japan. [4]Present address: Max Planck Institute for Solid State Research, Heisenbergstrasse 1, 70569 Stuttgart, Germany. [5]Present address: Physics Institute, University of Stuttgart, Pfaffenwaldring 57, D-70569 Stuttgart, Germany. [6]Present address: Department of Advanced Materials Science, University of Tokyo, Kashiwanoha 5-1-5, Kashiwa 277-8561 Chiba, Japan. ✉e-mail: kanoda@ap.t.u-tokyo.ac.jp

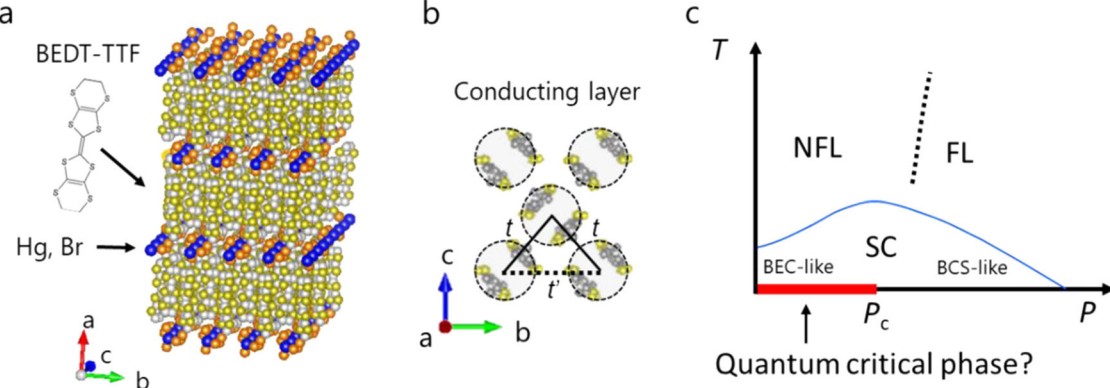

**Fig. 1 | Crystal structure and schematic phase diagram of κ-HgBr. a** Layered crystal structure of κ-HgBr. The orange and blue spheres indicate Br and Hg ions in the insulating layers, respectively. **b** In-plane molecular arrangement in the conducting layer of κ-HgBr. The BEDT-TTF molecules form dimers (circled by dotted lines), which construct an isosceles triangular lattice, characterised by two kinds of transfer integrals of $t$ and $t'$ between the adjacent antibonding dimer orbitals; the

ratio, $t'/t$, is 1.02 according to the molecular orbital calculations (see Supplementary Note 1). **c** Schematic pressure-temperature phase diagram of κ-HgBr drawn with reference to the previous studies[16,19,20]. The NFL, FL, SC, and $P_c$ stand for non-Fermi liquid, Fermi liquid, superconductivity, and critical or crossover pressure between NFL and FL, respectively. The red bold line indicates the possible critical phase inferred from ref. 16,19,20.

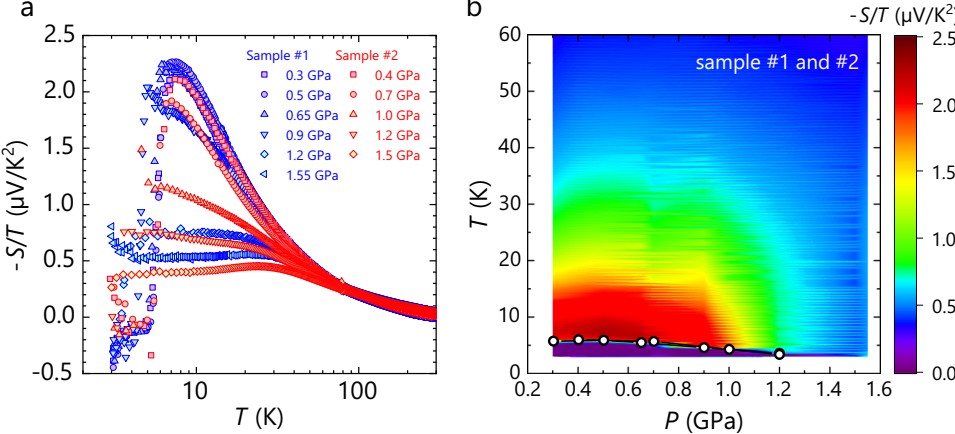

**Fig. 2 | Temperature and pressure profiles of −$S/T$ in κ-HgBr. a** Temperature dependence of the Seebeck coefficient divided by temperature, −$S/T$, in the samples #1 and #2, whose data are distinguished by different colours. The essential features of the results coincide with each other. Thin straight lines indicate the dependence of $S/T \propto \ln(1/T)$. **b** Contour plot of the −$S/T$ values shown in Fig. 2a in the

pressure-temperature plane. The open circles indicate the superconducting transition temperature $T_c$ determined from −$S/T$ (see Fig. 4 and Supplementary Fig. 3 for the definition of $T_c$). The separate plots for each sample are shown in Supplementary Fig. 1.

material, κ-(BEDT-TTF)$_2$Cu$_2$(CN)$_3$, thus suggesting that κ-HgBr hosts a doped QSL[15]. The electronic nature of κ-HgBr is indicated to alter by pressure. The Hall coefficient behaves such that charge carriers are only the doped holes at low pressures due to a strong correlation prohibiting double occupancy but is recovered to full band carriers at high pressures[16,17]. This appears to be a pressure equivalence of the doping-driven $p$ to $1+p$ crossover with $p$ a doping content in cuprates[18]. In resistivity, the non-Fermi-liquid persists up to ~0.4–0.5 GPa and crosses over or transitions to a Fermi liquid[16,19,20], as theoretically suggested[21]. At low temperatures, superconductivity occurs whose transition temperature, $T_c$, shows dome-like pressure dependence and whose nature changes from BEC-like to BCS condensate[19]. The schematic phase diagram is shown in Fig. 1c.

The quantum criticality in a phase instead of at a point and its possible relevance to QSL is an issue of profound significance. The present work aims to verify the quantum critical nature of the electronic state in κ-HgBr with pressure variation through the thermoelectric effect which is very susceptible to quantum criticality, exploiting the highly compressible feature of organic crystal[22]. Here,

we report our observation of the thermoelectric signature of quantum criticality in the doped QSL phase and its possible relevance to superconductivity.

## Results and discussion

Figure 2a shows the temperature dependence of the Seebeck coefficients divided by temperature, −$S/T$, under several pressures. Two separate measurements on different κ-HgBr samples (distinguished by different colours in Fig. 2a) give nearly coinciding results. To view the overall profile of −$S/T$ in the pressure-temperature plane, we display the values with a range of colours in Fig. 2b (separate plots for each sample are shown in Supplementary Fig. 1). −$S/T$ behaves similarly at every pressure at high temperatures above 30–40 K but, below that, shows strong pressure dependence with fan-shaped dispersion; −$S/T$ is highly enhanced at low pressures well below 1 GPa whereas it is progressively reduced with increasing pressure. A sudden decrease in −$S/T$ at low temperatures is due to the superconducting transition, as described in detail later.

At high pressures above 1 GPa, where the electron correlation is weakened, −$S/T$ is constant at low temperatures, being consistent with

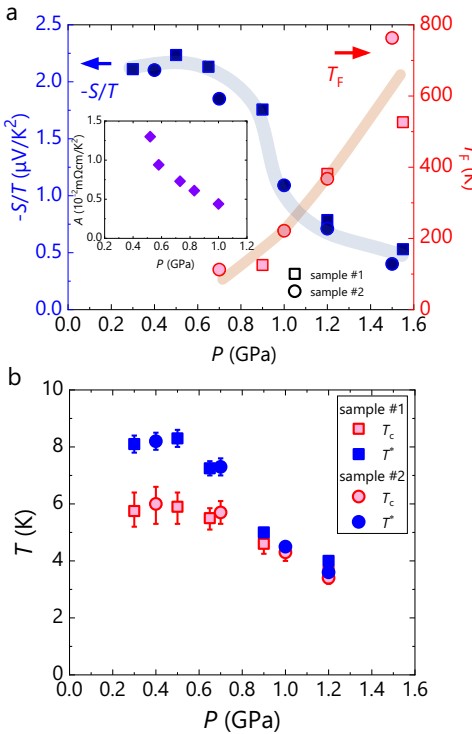

**Fig. 3 | Pressure dependences of −S/T and superconducting transition temperature in κ-HgBr. a** Pressure dependences of the −S/T value at 8 K, just above $T_c$, and the Fermi temperature, $T_F$. The square and circle markers correspond to samples #1 and #2, respectively. The blue and red markers are the −S/T and $T_F$ values, respectively. Inset shows the pressure dependence of the coefficient, A, in the fit of the form, $\rho = \rho_0 + AT^2$, to the separately measured resistivity. **b** Pressure dependences of the superconducting transition temperature, $T_c$, and its onset, $T^*$. The error bars of $T_c$ indicate the widths of the bulk superconducting transition. $T^*$ is defined as the temperature at which −S/T starts to deviate from the normal-state behaviour. The definitions of $T_c$, its error bar and $T^*$ are described in Supplementary Fig. 3.

the Fermi-liquid behaviour observed in resistivity[16,19,20] (see also Supplementary Fig. 2). For Fermi liquids, S is expected to follow the formula[23],

$$\frac{S}{T} = \pm \frac{\pi^2}{3}(1+\lambda)\frac{k_B}{e}\frac{1}{T_F},\qquad(1)$$

where $T_F$ is the Fermi temperature and $\lambda$ is a parameter related to the energy dependence of relaxation time, e.g., $\lambda = 0$ in the case of constant (energy-independent) mean free path. The −S/T values at low temperatures are 0.4–0.53 μV/K² at 1.50-1.55 GPa, -0.75 μV/K² at 1.2 GPa, and -1.1 μV/K² at 1.0 GPa and, assuming $\lambda = 0$, these −S/T values yield $T_F$ = 530–710, ~380 and ~260 K at 1.50-1.55, 1.2 and 1.0 GPa, respectively. The S/T data for 0.9 and 0.7 GPa does not saturate in the normal state but shows appreciable deviations from the logarithmic T-dependence discussed below, which may be a sign of the Fermi liquid at lower temperatures. As the superconductivity at these pressures can be suppressed by magnetic fields of several Tesla, we deduced the Fermi temperatures from the extrapolations of S/T under magnetic fields suppressing superconductivity to zero kelvin in linear scales; ~2.3 μV/K² at 0.9 GPa, ~2.5 μV/K² at 0.7 GPa. S/T appears to saturate on approaching $T_c$ even below 0.5 GPa. Our previous Nernst-effect measurements suggest the enhanced preformation of the Cooper pairs, particularly below 0.5 GPa[19]. As the saturation of S/T from the logarithmic T-dependence may be due to the superconducting fluctuations, the low-temperature limit of −S/T in the normal state

should be larger than the peak value of ~2.2 μV/K² at 8 K shown in Fig. 3a; namely, the Fermi temperature is lower than ~100 K. $T_F$ decreases with pressure toward zero around 0.5 GPa (Fig. 3a), which is very probably ascribable to the progressive renormalisation of the Coulomb interaction. Concomitantly, the temperature dependence of −S/T starts to deviate from the Fermi liquid behaviour of S/T = constant.

With further decreasing pressure below 1 GPa, the temperature dependence of −S/T deviates appreciably from the Fermi liquid behaviour, and −S/T continues to increase on cooling until superconductivity sets in at $T_c$. The low-temperature value just above $T_c$ reaches the values over 2.0 μV/K² at 0.5–0.65 GPa and levels off at lower pressures (Fig. 3a), where non-Fermi liquid behaviour of resistivity is observed[16,19,20]. The temperature profile of −S/T in the low-pressure region is roughly linear in Fig. 2a, meaning S/T∝ln(1/T). Such temperature dependence of the Seebeck coefficient is observed as a signature of quantum criticality in strongly correlated systems such as cuprates[24–27], iron pnictide[28–30], heavy fermion[31–36] and cobalt oxides[37], and intensively studied theoretically[38–41]. To be quantitative, the temperature dependences of −S/T under pressures below 0.5 GPa were fitted by the form of S/T = γ′ln($T_0$/T), where $T_0$ is a parameter of the energy scale of quantum critical fluctuations[38]. The fitting yields $T_0$ = 50–60 K, which is compared to $T_0$-170 K for Nd-LSCO and $T_0$-3 K for YbRh₂Si₂ (ref. 24,31); these values appear consistent with their relative sizes of bandwidths of organic conductors, cuprates and heavy electron systems.

Thus, the present observation provides evidence for quantum criticality in the low-pressure region in κ-HgBr. A distinctive feature from the conventional cases is that it is extended in a finite pressure range, namely, in a "critical region" instead of a "critical point". In the present case, both the magnitude and logarithmic behaviour of −S/T maintain unchanged below 0.5–0.65 GPa as seen in Fig. 2a. One should be cautious about a possible case that the fanning out toward a finite temperature of quantum criticality originating from a single point is hidden by the superconducting phase. Such a case is argued, e.g., for Ba(Fe₁₋ₓCoₓ)₂As₂, in which the magnitude of S/T and the coefficient of its logarithmic temperature dependence above $T_c$ exhibit cusp-like anomalies at a doping level and an SDW transition appears to vanish at that doping level, suggesting a symmetry-breaking quantum phase transition[29]. These features provide good reasons to suggest the existence of a quantum critical point in Ba(Fe₁₋ₓCoₓ)₂As₂. κ-HgBr, however, shows no appreciable anomalies in the magnitude and temperature dependence in S/T in the quantum critical pressure region (Figs. 2a, b, and 3a), and no symmetry-breaking phase transition in the pressure range studied. Thus, the present experimental data have no indications supporting a critical point. Furthermore, frustration-induced non-ordered phases as in the present system are theoretically suggested to likely host quantum critical phases extended in a finite range of control parameter[8,9]. Considering that other materials with spin frustration are argued to host such a phase[4–7], spin frustration would be a key to the stabilisation of the quantum critical phase, as suggested theoretically[8,9].

It is noted that the enhanced S/T values are not sharply suppressed upon the crossover from the non-Fermi liquid to the Fermi liquid at -0.5 GPa. We consider this as a possible manifestation of the strong electron correlation in the marginal Fermi liquid nearby a non-Fermi liquid. The coefficient A in the temperature dependence of resistivity, $\rho(T) = \rho_0 + AT^2$, in the Fermi-liquid regime is a measure of correlation strength or quasi-particle scattering rate due to Coulomb interactions. The pressure dependence of A measured with a separate sample is displayed in the inset of Fig. 3a, which exhibits its remarkable increase well before entering the non-Fermi liquid regime (see Supplementary Fig. 5 for the resistivity data). However, the seeming discrepancy between the non-saturated S/T and the quadratic resistivity under pressures of 0.5–1.0 GPa requires further consideration, which

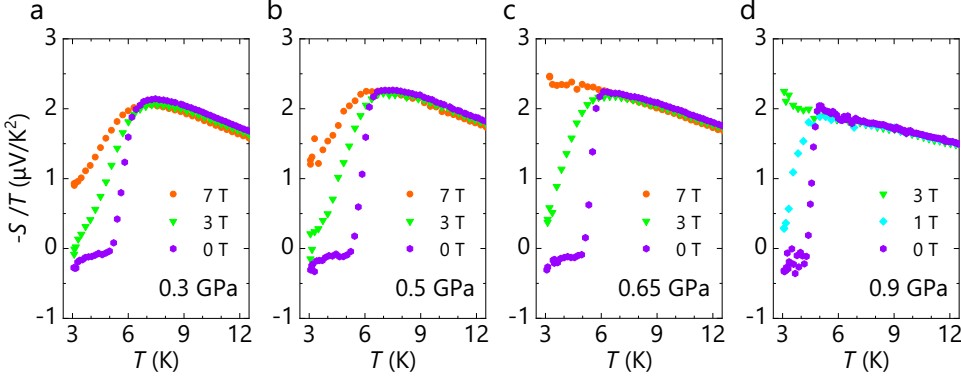

**Fig. 4 | Temperature dependence of −*S/T* at zero and applied magnetic fields. a–d** Temperature dependence of −*S/T* (for sample #1) under the pressures of 0.3 (**a**), 0.5 (**b**), 0.65 (**c**) and 0.9 GPa (**d**). Magnetic field was applied perpendicular to the conducting plane.

invokes different scattering profiles in the temperature-driven and force-driven electron diffusions. The Seebeck coefficient is expressed by $S(T) \propto (\rho(T)/T) \int \sigma(\varepsilon,T)(\varepsilon - \varepsilon_F)\{-f'(\varepsilon,T)\}\mathrm{d}\varepsilon$ with the conductivity, $\sigma(\varepsilon,T)$, and the energy derivative of the Fermi-Dirac function, $f'(\varepsilon,T)$. We note that $(\varepsilon - \varepsilon_F)\{-f'(\varepsilon)\}$ takes a maximum away from the Fermi energy $\varepsilon_F$ by $\sim 1.5 k_B T$, meaning that the relevant energy range for $S$ not only thermally broadens but progressively shifts to higher energies as the temperature is raised. This energy shift should cause significant effects beyond an effective temperature rise in correspondence with the behaviour of electrical resistivity which shows the $T^2$ dependence at low temperatures and deviates towards lower powers of temperature above 15–20 K (see Supplementary Fig. 5a and ref. [19]). Sample dependence and uncertainty in pressure determination in different runs may be additional origins of the discrepancy.

In many cases, the quantum critical logarithmic-in-temperature evolution of −*S/T* appears in the vicinity of magnetic transitions[38]. In κ-HgBr, enhanced spin fluctuations are suggested by NMR studies[42] and thus likely involved in the enhanced *S/T* albeit in a different way from the magnetic quantum criticality because of no magnetic order in κ-HgBr. It is known that $S$ is empirically well expressed by $S \sim C/ne$ in $T \to 0$ limit, where phonon contribution is negligible; $C$ is specific heat and $n$ is the density of charge carriers with charge $e$. Thus, $S$ is roughly an entropy per charge carrier[43]. As indicated by the Hall coefficient and resistivity, the nature of charge carriers is changed with decreasing pressure from the band quasiparticles with Fermi liquidity to emergent holes (that should be called holons) with non-Fermi liquidity, which originates from the prohibition of double occupancy—akin to the Mott localisation in a half-filled system. This drastic change of the carrier nature with no magnetic symmetry breaking should affect the enhanced *S/T*. The emergent holons may have extraordinary charge fluctuations that are entangled with a QSL having large entropy[44]. The QSL that possibly extends in the low-pressure range is considered a key to the quantum criticality persisting in a finite parameter range, not at a point[8,9]. As *S/T* probes the effective mass or density of states through the relation, $S \sim C/ne$, the coefficient $|\gamma'|$ of the logarithmic *T*-dependence is a possible measure of the strength of the quantum criticality, which appears in the specific heat or effective mass[38]. The reported values of $|\gamma'|$ are 0.01–0.05 μV/K² for electron-doped cuprates[27], 0.05–0.16 μV/K² for hole-doped cuprates[24–26], 0.3–0.9 μV/K² for ion pnictides (Ba(Fe$_{1-x}$Co$_x$)$_2$As$_2$) (ref. [29]), and 2.3, 4.5, 6.2 μV/K² for heavy electron systems (UCoGe, YbRh$_2$Si$_2$, and CeCu$_{5.9}$Au$_{0.1}$, respectively)[31–33] (see Supplementary Table 1 for $|\gamma'|$ values of other materials). Thus, the present $|\gamma'|$ value for κ-HgBr, -1.2 μV/K², is situated in between the values of the cuprates and heavy electrons, reflecting the density of states or inverse of the Fermi energy.

Remarkably, the superconducting transition temperature $T_c$ is well correlated with the low-temperature values of the logarithmically

enhanced −*S/T*, as seen in Figs. 3a, b (see Supplementary Figs. 3 and 4 for definitions of $T_c$ and onset $T^*$, and the pressure dependences of $|\gamma'|$ and $T_c$, respectively). Given that $|\gamma'|$ is an indicator of the strength of critical fluctuation[38], the correlation suggests that the critical fluctuations mediate or facilitate the electron pairing. Such correlation is also found in cuprates and iron pnictides as well[27,29]. Figure 4 shows the low-temperature behaviour of −*S/T* upon superconducting transition under zero and applied magnetic fields perpendicular to the conducting layers. −*S/T* vanishes in the superconducting state. The transition is quite sharp at high pressures; at lower pressures, however, it becomes rounded with an onset well prior to the bulk transition (Fig. 3b), reserving the possibility of enhanced superconducting fluctuations. Figure 4 also shows that the superconductivity is entirely destroyed by a field of 3 T under 0.9 GPa whereas, under 0.3 GPa, it survives even at a field of 7 T albeit partially very probably as a vortex liquid state. These superconductive features are fully consistent with the previously revealed BEC-to-BCS crossover associated with a non-Fermi liquid to a Fermi liquid crossover in κ-HgBr (ref. [19]). In the BEC-like regime at low pressures, the superconductivity shows enhanced fluctuations with preformed Cooper pairs and is robust to the magnetic field with forming a vortex liquid state[19].

The nature of the possible underlying spin liquid in κ-HgBr is of profound interest and the present observation potentially gives a clue to it. However, since the present Seebeck effect mainly comes from mobile charge carriers, not directly from spins, it is not straightforward to discuss the underlying spin-liquid nature from the present observation. Moreover, the Hubbard model with the intermediate strength of interaction, which is appropriate to κ-HgBr, suggests the complexity of competing phases on a triangular lattice[45]. Nevertheless, it is suggestive that a spin liquid is theoretically generated from superconductivity by Gutzwiller projection, namely, removing double occupancies and the symmetries of the two phases are mutually correspondent like d-wave superconductivity vs. Dirac (nodal) spin liquid or chiral superconductivity vs. chiral spin liquid[46]. Therefore, the symmetry of superconductivity emerging from a spin liquid under pressure is quite informative because the pressure-induced Mott transition, which prohibits double occupancies, can be the experimental realisation of the Gutzwiller projection. The superconductivity that appears by pressurising the spin-liquid candidate κ-(ET)$_2$Cu$_2$(CN)$_3$, the undoped analogue of κ-HgBr, is shown to be nodal by NMR experiments[47]. A Dirac spin liquid[48] appears likely. Furthermore, a μSR study of κ-HgBr showed the absence of time-reversal symmetry breaking in the superconducting state[49], implying less likeliness of a chiral spin liquid. Anyway, the nature of the underlying spin liquid is an issue of future investigation.

The present work is the first thermoelectric investigation of a doped spin-liquid candidate. The thermoelectric effect probes the

entropy transport by charge carriers, which are influenced by magnetic background and superconductivity if any, and therefore includes information on the surroundings of the doped holes. The logarithmic Seebeck enhancement observed at low pressures signifies that charge carriers that travel, avoiding double occupancies, in the sea of spin liquid suffer from quantum critical fluctuations in charge and/or spin degrees of freedom. It is emphasised that the quantum critical state resides as a phase, not at a point. As pressure is increased, the logarithmic enhancement is suppressed and crosses over to the conventional metallic behaviour, indicating that the doped spin liquid crosses over to a Fermi liquid by reducing the Coulomb interactions. The correlation between the logarithmic Seebeck enhancement and superconductivity suggests that the anomalous quantum critical fluctuations favour the BEC-like electron pairing. It is an issue of further investigation whether spin or charge fluctuations or both mediate the Cooper pairing.

## Methods

Single crystals of κ-HgBr were grown in the standard electrochemical method. For pressurisation, we used a clamp-type piston-cylinder cell made of CuBe/NiCrAl and Daphne oil 7373 as a pressure-transmitting media. Daphne oil 7373 solidifies on cooling so that the clumped pressure gradually decreases by 0.15–0.2 GPa as the temperature decreases from 300 K to 50 K and then takes a nearly constant value at lower temperatures[50]. To know the internal pressure in the piston-cylinder cell, we used $T_c$ of a Sn flake that was mounted in the cell. The pressure values quoted in this article are the internal pressures thus estimated.

Thermoelectric effect was measured with a conventional experimental platform where two Cu-plates with the Cernox thermometers attached on both and a heater attached on one plate are bridged by a κ-HgBr crystal. The thermometers were calibrated at each pressure using a reference thermometer. The heater generated a temperature difference, $\Delta T$, between the two Cu-plates, which was maintained $<T/10$ throughout the experiments. With measuring thermoelectric potential difference, $\Delta V$, between the plates under the temperature difference, $\Delta T$, the Seebeck coefficient is defined by $S = \Delta V/\Delta T$. In the present experiment, temperature gradient was applied along the c-axis in the conduction plane (Fig. 1b).

The rapid cooling is often detrimental to organic conductors because it may cause crystal cracking and/or conformational disorder of terminal ethylene groups in BEDT-TTF. To minimise these possible faults, we cooled the sample at rates slower than 0.5 K/min. Even with such cautious cooling process, the cracking in κ-HgBr crystal was not avoided at ambient pressure. Thus, the present experiments were performed under pressures, where the sample was free from such a problem.

## Data availability

The data that support the discussion and conclusion in the present paper are all presented in the main manuscript and Supplementary Information online. Additional data, e.g., numerical values, are available from the corresponding author upon reasonable request.

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

## Acknowledgements

We thank H. Oike and M. Ogata for the fruitful discussion. This work was supported by the Japan Society for the Promotion of Science (JSPS) under Grant Numbers 18H05225 (K.K.), 19H01846 (K.M.), 20K20894 (K.M.), 20KK0060 (K.M. and K.K.) and 21K18144 (K.K.), and by JST SPRING under Grant Number JPMJSP2108 (K.W.). Most parts of this work were performed using facilities of the Cryogenic Research Center, at the University of Tokyo.

## Author contributions

K.K. designed the project. H.T. prepared samples. K.W., Y.S., T.F., and K.M. performed experiments and analysed as well as interpreted data with the help of K.K. K.W., and K.K. wrote the manuscript with the input from all authors.

## Competing interests

The authors declare no competing interests.
