## [Peer Review File · Nature Communications]

REVIEWER COMMENTS

Reviewer #1 (Remarks to the Author):

The manuscript “Thermoelectric signature of quantum critical phase in a doped spin liquid candidate” by K. Wakamatsu et al. is a study of the hydrostatic pressure dependence of the thermoelectric response in κ -(BEDT-TTF)₄Hg_{2.89}Br₈ (shortly: κ -HgBr). The main result is the observation of the low temperature logarithmic divergence of the normalised Seebeck coefficient (S/T) for the pressure range $p \sim 0.3 - 1$ GPa. This led the authors to conclude that they detected the quantum criticality in a form of the quantum critical phase in κ -HgBr. The authors also claim that the thermoelectric response is consistent with the BEC-BCS scenario.

I find the topic of the paper attractive and important. There were some recent reports about emergence of quantum critical phases in CePdAl [H. Zhao et al., Nat. Phys. 15, 1261 (2019)], YbRh₂Si₂ [S. Friedemann et al., Nat. Phys. 5, 465 (2009)] and β -YbAlB₄ [T. Tomita et al., Science 349, 506 (2015)] but the number of examples is very limited. The phenomenon is not fully understood thus a discovery of another material exhibiting the quantum critical phase would certainly be worth publishing. However, I do not think that the manuscript in its current form is convincing enough. Here is a list of my questions and comments:

1. κ -HgBr becomes a superconductor in the range of pressures, for which the authors claim presence of the quantum critical phase. However, it was shown that also in the iron-based superconductor Ba(Fe_{1-x}Cox)₂As₂ [S. Arsenijevic et al., Phys. Rev. B 87, 224508 (2013)] the logarithmic divergence of S/T occurs over a wide range of doping. This is supposedly due to the presence of the quantum critical point under the superconducting dome. Why would the case of κ -HgBr be different?
2. The pressure dependence of the parameter A (shown in inset in Fig. 3a) seems to have a singularity at $p \sim 0.5$ GPa. Is it not related to the quantum critical point at this pressure?
3. The thermoelectric properties of κ -HgBr are said to be in line with the BEC-BCS scenario and also that quantum critical fluctuations favour the BEC-like electron coupling. The reasoning behind these claims is not clear to me. Was the enhancement of superconducting fluctuations also observed in the resistivity?
4. The phase diagrams presented in Fig. 2c and Fig. S1 are aesthetically attractive but they are based on a very limited number of experimental data. For example, in Fig. 2c, the entire important section between $p \sim 0.9$ and 1.2 GPa appears to be a sort of “artistic imagination”. I would prefer the plots to show real data, even if that makes them less catchy. Perhaps the authors could consider combining results from samples #1 and #2 in a single plot.
5. The authors write that S/T diverges as $\sim \ln(T)$ but in fact it is rather $\sim \ln(1/T)$. Analogously, the function used to determine the T_0 parameter should be corrected.
6. What do the authors mean by “quasiparticle dumping rate” (l. 100)?

7. The authors write that S may be expressed by $S \sim C/ne$ at temperatures (l. 107). What temperatures?
8. It is suggested that the quantum criticality is strongly coupled to the thermoelectric effect (l.119). Could the authors be more specific?
9. Finally, I would like to point out that the text is not clearly written and requires thorough stylistic corrections.

In conclusion, I cannot recommend publication of the manuscript in its current form. I believe it contains some interesting data, but they are presented and interpreted in a way not meeting the Nature Communications standards. Perhaps the authors will be able to prepare and resubmit the thoroughly revised version of their work.

Reviewer #2 (Remarks to the Author):

Wakamatsu et al. Report on a study of the evolution of the Seebeck coefficient in kappa-ET organic conductor, which has a superconducting round state. With its triangular lattice, this solid is a doped spin-liquid candidate and worthy of study. The results are new, the quality of data is high, and the scientific analysis is sound. I recommend publication, provided that the authors address the following issues:

1. The authors use equation 1 the data shown in Fig. 2 to extract the Fermi temperature and its pressure dependence, shown in Fig. 3. It is not clear to me what criterion they use to stop doing this below 1 GPa.
2. The amplitude of the Seebeck coefficient is set by entropy per mobile carrier. At low pressures, when S/T is not flat and increases with decreasing temperature, its amplitude is set by the degeneracy temperature of electrons (which keeps rising as the system is cooled down). Equation 1 remains valid even when $p=0.4$ GPa at $T \sim T_c=7$ K, the Fermi temperature extracted from $S/T \sim 2.2 \mu\text{V}/\text{K}^2$ is about 100 K. This is more than one order of magnitude larger than the critical temperature. The authors should inform their readers that even if a $T_c/T_F \sim 0.1$ is remarkably large, the degeneracy temperature of Fermions is much higher than the onset of superconductivity. In this sense even at 0.3 GPa, the system is NOT a BEC.
3. It would be a good idea to compare the peak S/T obtained here, with other cases of quantum criticality: CeCoIn5 [$S/T > 6 \mu\text{V}/\text{K}^2$, PRL PRL 99, 147005 (2007)], YbRh2Si2 [$S/T > 8 \mu\text{V}/\text{K}^2$, PRL PRL 109, 156405 (2012)], and CeCu5.9Au0.1 [$S/T > 30 \mu\text{V}/\text{K}^2$, Physica B 259-261, 380 (1999) and Physica B 281-282, 359 (2000)]. Note that the Fermi temperature in these heavy-fermion quantum critical systems, becomes as small as a few Kelvins.

4. Fig. contains an inset showing the pressure dependence of A, the prefactor of T-square resistivity. Where is the resistivity data to see how A has been extracted and what is the temperature window of T-square resistivity?
5. If the system is not a Fermi liquid below 1 GPa, how come that its resistivity follows a T-square temperature dependence?
6. Is the absolute value of A reliable (despite cracks appearing in such fragile samples under pressure)? If yes, the magnitude of A can also be a guide to the Fermi temperature [See for example Fig4b in Nature Commun. 11: 3846 (2020)].

Reviewer #3 (Remarks to the Author):

The authors performed thermoelectric measurements on the material κ -HgBr and demonstrated that the Seebeck effect exhibits features of quantum criticality. To this end, the Seebeck coefficient is studied as both a function of temperature and pressure. Interestingly, the quantum critical behavior persists for a finite region of the phase diagram. Hence, the authors propose that the criticality does not stem from a single critical point but rather an extended critical phase. Since transport in this material is essentially taking place in two dimensions, their finding would constitute the discovery of a critical phase in two dimensions, which is a rather exciting prospect. Critical phases are well known in one spatial dimension, but agreement on whether they could exist in two dimensions has currently not been achieved. The results are presented in a clear way and the paper is written in a very accessible style. I, therefore, think that the results are highly topical and deserve publication in some form. However, I would like the authors to take into account some of the following questions and remarks before I can make a final decision.

1) The authors claim in the abstract that this material is currently the only candidate of a doped spin liquid known. I think this is a too bold statement to make at the present moment. There are several novel experimental platforms of doped kagome antiferromagnets and Kitaev systems that are under investigation and where quantum spin liquids are seriously considered. I suggest removing the statement that this material is the only candidate for a doped spin liquid.

2) The authors should discuss theoretical proposals for critical phases in more detail. The most prominent critical spin liquid state which constitutes a phase in two dimensions is the Dirac spin liquid. While there are several theoretical proposals that this state can be realized in triangular geometries, its existence is not certain. In fact, it is as of today not known whether such a state can actually be a stable phase, e.g.

<https://journals.aps.org/prl/abstract/10.1103/PhysRevLett.123.207203>.

3) Recent theoretical proposals have studied the effect of hole doping a spin liquid in the triangular lattice Hubbard model and its thermodynamics, e.g.

<https://journals.aps.org/prl/abstract/10.1103/PhysRevLett.125.157002> and

<https://journals.aps.org/prx/abstract/10.1103/PhysRevX.11.041013>. The manuscript would benefit from discussing, whether the author's findings could be explained by such theoretical models.

[Reply to Reviewer #1]

Comment #1-0

The manuscript “Thermoelectric signature of quantum critical phase in a doped spin liquid candidate” by K. Wakamatsu et al. is a study of the hydrostatic pressure dependence of the thermoelectric response in κ -(BEDT-TTF) $_4$ Hg $_2$.89Br $_8$ (shortly: κ -HgBr). The main result is the observation of the low temperature logarithmic divergence of the normalized Seebeck coefficient (S/T) for the pressure range $p \sim 0.3 - 1$ GPa. This led the authors to conclude that they detected the quantum criticality in a form of the quantum critical phase in κ -HgBr. The authors also claim that the thermoelectric response is consistent with the BEC-BCS scenario. I find the topic of the paper attractive and important. There were some recent reports about emergence of quantum critical phases in CePdAl [H. Zhao et al., *Nat. Phys.* 15, 1261 (2019)], YbRh $_2$ Si $_2$ [S. Friedemann et al., *Nat. Phys.* 5, 465 (2009)] and β -YbAlB $_4$ [T. Tomita et al., *Science* 349, 506 (2015)] but the number of examples is very limited. The phenomenon is not fully understood thus a discovery of another material exhibiting the quantum critical phase would certainly be worth publishing. However, I do not think that the manuscript in its current form is convincing enough. Here is a list of my questions and comments:

Reply #1-0

We are grateful to Reviewer #1 for reading our manuscript thoroughly and giving us enlightening comments for improving the MS. Having the comments, we scrutinized the previous MS and then revised it, fully taking the comments into consideration.

Below, we reply point by point to all the comments and describe the revised parts, which are underlined in the following replies and colored in the MS.

We included the suggested references and a related one (Ref.6) in the revised MS:

Ref.4) Friedemann, S., Westerkamp, T., Brando, M., Oeschler, N., Wirth, S., Gegenwart, P., Krellner, C., Geibel, C. & Steglich, F. Detaching the antiferromagnetic quantum critical point from the Fermi-surface reconstruction in YbRh $_2$ Si $_2$. *Nat. Phys.* **5**, 465-469 (2009).

Ref.5) Tomita, T., Kuga, K., Uwatoko, Y., Coleman, P. & Nakatsuji, S. Strange metal without magnetic criticality. *Science* **349**, 506 - 509 (2015).

Ref. 6) Machida, Y. et al. Thermoelectric Response Near a Quantum Critical Point of β -YbAlB $_4$ and YbRh $_2$ Si $_2$: A Comparative Study. *Phys. Rev. Lett.* **109**, 156405 (2012).

Ref.7) Zhao, H. et al. Quantum-critical phase from frustrated magnetism in a strongly correlated metal. *Nat. Phys.* **15**, 1261–1266 (2019).

Comment #1-1

1. κ -HgBr becomes a superconductor in the range of pressures, for which the authors claim presence of the quantum critical phase. However, it was shown that also in the iron-based superconductor Ba(Fe $_{1-x}$ Co $_x$) $_2$ As $_2$ [S. Arsenijevic et al., *Phys. Rev. B* 87, 224508 (2013)] the logarithmic divergence of S/T occurs over a wide range of doping. This is supposedly due to the presence of the quantum critical point under the superconducting dome. Why would the case of κ -HgBr be different?

Reply #1-1

In Ba(Fe $_{1-x}$ Co $_x$) $_2$ As $_2$, the quantum criticality is argued to spring from a transition “point” from a SDW state to a paramagnetic state with increasing x , as the reviewer says. However, the present system, κ -HgBr, does not show magnetic order in the entire pressure range studied because the geometrical frustration in the triangular lattice prohibits magnetic order.

In the absence of magnetic order, a Fermi liquid at high pressures changes to a non-Fermi liquid with decreasing pressure and the non-Fermi liquid state extends in a low-pressure range as a “phase” not at a point. This non-Fermi liquid “phase” was shown to be in a quantum critical state by the present work (regarding to the more extended quantum critical region than the non-Fermi liquid region, please see Reply #1-2). In the low-pressure non-Fermi liquid phase, only the doped holes are mobile due to strong prohibition of double occupancies as indicated by the Hall coefficient measurements (Ref. 16) and nearly the same magnetism as in the spin-liquid candidate, κ -(ET)₂Cu₂(CN)₃, was observed (Ref.15). Therefore, the quantum criticality extended in a finite range of pressure is associated with a doped spin-liquid phase. The existence of quantum criticality in a finite range of parameter is theoretically rationalized for systems with geometrical frustration, which prohibits magnetic orders. In short, the magnetically ordered phase in Ba(Fe_{1-x}Co_x)₂As₂ is replaced by non-Fermi liquid phase with quantum spin liquidity, that exhibits quantum criticality. In the original MS, we explained the above in the paragraph starting from “In many cases, the quantum critical logarithmic-in-temperature evolution of $-S/T$ appears in the vicinity of magnetic transitions³⁸. In κ -HgBr, ……………”. To emphasize the above discussion, we added the following sentence to this paragraph.

[In 9th paragraph] “The QSL that possibly extends in the low-pressure range is considered as a key to the quantum criticality persisting in a finite parameter range, not at a point.”

Comment #1-2

2. The pressure dependence of the parameter A (shown in inset in Fig. 3a) seems to have a singularity at $p \sim 0.5$ GPa. Is it not related to the quantum critical point at this pressure?

Reply #1-2 (this is common to Comment/Reply #2-5 below)

Figs.2a and 2b show that the logarithmic T -dependence of S/T appears in a low-pressure region but doesn't look like it's coming out of one point. This feature has an explanation as theoretically supported (please see Reply #1-1). In resistivity, the non-Fermi liquid behavior at low pressures crosses over at $p \sim 0.5$ GPa to the Fermi liquid one characterized by the value of A . However, S/T keeps increasing on cooling under pressures up to $p \sim 1$ GPa as if the quantum critical behavior persists up to $p \sim 1$ GPa. Although the seemingly inconsistent behaviours between resistivity and S/T in 0.5-1.0 GPa should be further investigated, the conceivable reasons are as follows.

As seen in Fig. 2a in the revised MS (Figs. 2a and 2b in the previous MS are combined into the new Fig. 2a, following Reviewer#1's suggestion), $-S/T$ under 0.7 and 0.9 GPa shows downward deviation from the logarithmic dependence at low temperatures, suggesting possible Fermi liquid behavior well below 10 K in this pressure range. From the Boltzmann equation, $S(T) \propto \int \sigma(\varepsilon)(\varepsilon - \varepsilon_F)\{-f'(\varepsilon)\}d\varepsilon$ while the conductivity is expressed as $\sigma(T) \propto \int \sigma(\varepsilon)\{-f'(\varepsilon)\}d\varepsilon$, where $\sigma(\varepsilon)$ is the energy-dependent conductivity. We note that, while the factor of $-f'(\varepsilon)$ in $\sigma(T)$ gives thermal average at $\varepsilon = \varepsilon_F$, the factor of $(\varepsilon - \varepsilon_F)\{-f'(\varepsilon)\}$ in $S(T)$ roughly gives thermal average at an energy higher (by roughly $k_B T$) than ε_F . As seen in Fig. S5a, the resistivity deviates from the quadratic temperature dependence towards lower powers of temperatures around 15-20 K depending on pressure. This suggests that a possible origin of the discrepancy is the difference in the energy range probed by the Seebeck coefficient and resistivity. Additionally, we do not rule out the sample dependence and possible uncertainty in pressure determination. To describe these considerations, we added the following paragraph in the revised MS:

[In 8th paragraph] “Furthermore, we note that the energy range probed by S/T is higher by $\sim k_B T$ than the Fermi energy, at which the resistivity is determined. As mentioned above, the behavior of $-S/T$

under 0.7 and 0.9 GPa implies possible Fermi liquid behavior well below 10 K. In the pressure range of 0.5-1.0 GPa, the resistivity that exhibits the T^2 dependence at low temperatures starts to deviate towards lower powers of temperature around 15-20 K (see Fig. S5a in Supplementary Information). Thus, the seeming discrepancy between the logarithmic S/T and the quadratic resistivity under pressures of 0.5-1.0 GPa may be attributable to the difference in the energy ranges relevant to the two quantities. Sample dependence and uncertainty in pressure determination in different runs may be additional origins of the discrepancy.”

Comment #1-3

3. The thermoelectric properties of κ -HgBr are said to be in line with the BEC-BCS scenario and also that quantum critical fluctuations favour the BEC-like electron coupling. The reasoning behind these claims is not clear to me. Was the enhancement of superconducting fluctuations also observed in the resistivity?

Reply #1-3

In general, the superconducting “amplitude” fluctuations appear in resistivity in terms of the Aslamasov-Larkin and Maki-Thompson mechanisms. However, it is not clear how the superconducting “phase” fluctuations as in the BEC-like preformed Cooper pairs influence the resistivity. To the best of our knowledge, its effect on resistivity is suggested to be small theoretically [A] and experimentally [B]. Instead, the theory [A] predicts that the Nernst effect sensitively captures the phase fluctuations. As for the present system, although detailed resistivity study in the light of the superconducting fluctuations remains to be performed, the fluctuation effect appears prominently in the Nernst signals from well above T_c (Ref. 19), which is consistent with the above literatures.

Because the discussion on the paraconductivity is beyond the scope of the present work, we do not include it in the revised MS.

[A] Chen, Q., Stajic, J., Tan, S., & Levin, K., BCS-BEC crossover: From high temperature superconductors to ultracold superfluids. *Phys Rep.* **412**(1) 1 (2005).

[B] Nakagawa, Y., Kasahara, Y., Nomoto, T., Arita, R., Nojima, T., & Iwasa, Y., Gate-controlled BCS-BEC crossover in a two-dimensional superconductor. *Science* **372**, 190 (2021).

Comment #1-4

4. The phase diagrams presented in Fig. 2c and Fig. S1 are aesthetically attractive but they are based on a very limited number of experimental data. For example, in Fig. 2c, the entire important section between $p \sim 0.9$ and 1.2 GPa appears to be a sort of “artistic imagination”. I would prefer the plots to show real data, even if that makes them less catchy. Perhaps the authors could consider combining results from samples #1 and #2 in a single plot.

Reply #1-4

Following the suggestion, we combined the experimental data of samples #1 and 2 previously plotted separately in Figs. 2a and 2b to present all of them in one figure (new Fig. 2a) and make a combined contour plot for all of the data (new Fig.2b, which is duplicated below) in the revised MS. The sample dependence of the results may be more easily recognizable by the separate data plots; so we show the separate plots in Supplementary Information. Accordingly, the relevant descriptions of Figure 2 are modified as follows:

[In 5th paragraph] “Figure 2a shows the temperature dependence of the Seebeck coefficients divided by temperature, $-S/T$, under several pressures. Two separate measurements on different κ -HgBr samples (distinguished by different colours in Fig.2a) give nearly coinciding results. To view the

overall profile of $-S/T$ in the pressure-temperature plane, we display the values with a range of colours in Fig. 2b (separate plots for each sample are shown in Supplementary Information)."

Comment #1-5

5. The authors write that S/T diverges as $\sim \ln(T)$ but in fact it is rather $\sim \ln(1/T)$. Analogously, the function used to determine the T_0 parameter should be corrected.

Reply #1-5

Following the suggestion, we rewrote the relevant parts with “ $S/T \propto \ln(1/T)$ ” in line 82 and with “ $S/T = \gamma \ln(T_0/T)$ ” in the 7th paragraph of the revised MS.

Comment #1-6

6. What do the authors mean by “quasiparticle dumping rate” (l. 100)?

Reply #1-6

We are sorry that dumping is the typo of damping. It is scattering rate; so, we replaced “quasi-particle damping rate” with “quasi-particle scattering rate due to Coulomb interactions” in the 8th paragraph of the revised MS.

Comment #1-7

7. The authors write that S may be expressed by $S \sim C/ne$ at temperatures (l. 107). What temperatures?

Reply #1-7

This is valid in $T \rightarrow 0$ limit, where phonon contribution is negligible. Thus, we rewrote “ S is

empirically well expressed by $S \sim C/ne$ in $T \rightarrow 0$ limit, where phonon contribution is negligible” in the 9th paragraph of the revised MS.

Comment #1-8

8. It is suggested that the quantum criticality is strongly coupled to the thermoelectric effect (l.119). Could the authors be more specific?

Reply #1-8

As seen in the expression, $S \sim C/ne$, the Seebeck coefficient, S , is directly connected to the specific heat, C ; so the S/T is supposed to be proportional to the effective mass or density of states and argued to probe the strength of the quantum critical fluctuations in Ref. 38, for example. To describe this, we modified the relevant descriptions in MS as follows:

[In 9th paragraph] “As S/T probes the effective mass or density of states through the relation, $S \sim C/ne$, the coefficient $|\gamma'|$ of the logarithmic T -dependence is a possible measure of the strength of the quantum criticality, which appear in the specific heat or effective mass³⁸. The reported values of $|\gamma'|$ are 0.01-0.05 $\mu\text{V}/\text{K}^2$ for electron-doped cuprates²⁷, 0.05-0.16 $\mu\text{V}/\text{K}^2$ for hole-doped cuprates²⁴⁻²⁶, 0.3-0.9 $\mu\text{V}/\text{K}^2$ for iron pnictides ($\text{Ba}(\text{Fe}_{1-x}\text{Co}_x)_2\text{As}_2$) (ref.²⁹), and 2.3, 4.5, 6.2 $\mu\text{V}/\text{K}^2$ for heavy electron systems (UCoGe , YbRh_2Si_2 , and $\text{CeCu}_{5.9}\text{Au}_{0.1}$, respectively)³¹⁻³³ (see Table S1 in Supplementary Information for $|\gamma'|$ values of other materials). Thus, the present $|\gamma'|$ value for $\kappa\text{-HgBr}$, $\sim 1.2 \mu\text{V}/\text{K}^2$, is situated in between the values of the cuprates and heavy electrons, reflecting the density of states or inverse of the Fermi energy.”

Comment #1-9

9. Finally, I would like to point out that the text is not clearly written and requires thorough stylistic corrections.

Reply #1-9

We reexamined MS and have tried to polish up the text. And, we consulted a English native about English sentences. The style of MS was corrected to meet the format of Nature Communications.

Comment #1-10

In conclusion, I cannot recommend publication of the manuscript in its current form. I believe it contains some interesting data, but they are presented and interpreted in a way not meeting the Nature Communications standards. Perhaps the authors will be able to prepare and resubmit the thoroughly revised version of their work.

Reply #1-10

We deeply thank the reviewer for helpful suggestions. We believe that the revised manuscript with the suggestions taken into account is much improved from the previous one and meets the Nature Communications standard.

[Reply to Reviewer #2]

Comment #2-0

Wakamatsu et al. Report on a study of the evolution of the Seebeck coefficient in kappa-ET organic conductor, which has a superconducting round state. With its triangular lattice, this solid is a doped spin-liquid candidate and worthy of study. The results are new, the quality of data is high, and the scientific analysis is sound. I recommend publication, provided that the authors address the following issues:

Reply #2-0

We are grateful to Reviewer #2 for reading our manuscript thoroughly and giving us enlightening comments for improving the MS. Having the comments, we scrutinized the previous MS and then revised it, fully taking the comments into consideration.

Below, we reply point by point to all the comments and describe the revised parts, which are underlined in the following replies and colored in the MS.

Comment #2-1

1. The authors use equation 1 the data shown in Fig. 2 to extract the Fermi temperature and its pressure dependence, shown in Fig. 3. It is not clear to me what criterion they use to stop doing this below 1 GPa.

Reply #2-1

We deduced the Fermi temperature only for the S/T data that show saturation (Fermi liquid behavior) at low temperatures. For pressures below 1 GPa, S/T does not saturate in the temperature range studied. Having the reviewer's suggestion, however, we reconsidered this criterion, paying attention to the fact that the data for 0.9 and 0.7 GPa show downward deviations from the logarithmic temperature dependence on cooling, which may be an indication of the Fermi liquid at lower temperatures. So, for these data, we deduced the Fermi temperatures from the extrapolations of S/T to zero kelvin in linear scales and additionally plotted the obtained values in Fig. 3a. Accordingly, the following explanation was added in the revised MS:

[In 6th paragraph] "The S/T data for 0.9 and 0.7 GPa, do not saturate in the normal state but show appreciable deviations from the logarithmic T -dependence discussed below, which may be a signature of the Fermi liquid at lower temperatures. As the superconductivity at these pressures can be suppressed by magnetic fields of several Tesla, we deduced the Fermi temperatures from the extrapolations of S/T under magnetic fields suppressing superconductivity to zero kelvin in linear scales; $\sim 2.3 \mu\text{V}/\text{K}^2$ at 0.9 GPa, $\sim 2.5 \mu\text{V}/\text{K}^2$ at 0.7 GPa"

Comment #2-2

2. The amplitude of the Seebeck coefficient is set by entropy per mobile carrier. At low pressures, when S/T is not flat and increases with decreasing temperature, its amplitude is set by the degeneracy temperature of electrons (which keeps rising as the system is cooled down). Equation 1 remains valid even when $p=0.4$ GPa at $T \sim T_c=7$ K, the Fermi temperature extracted from $S/T \sim 2.2 \mu\text{V}/\text{K}^2$ is about 100 K. This is more than one order of magnitude larger than the critical temperature. The authors should inform their readers that even if a $T_c/T_F \sim 0.1$ is remarkably large, the degeneracy temperature of Fermions is much higher than the onset of superconductivity. In this sense even at 0.3 GPa, the system is NOT a BEC.

Reply #2-2

We agree with the reviewer about mentioning the suggested information in MS. However, we think that we should be cautious about the following things. S/T certainly appears to saturate on approaching T_c even below 0.5 GPa. Our previous Nernst effect measurements (Ref. 19) indicate the enhancement of preformed Cooper pairs particularly under pressures below 0.5 GPa. Thus, the saturation tendency of S/T near T_c may be due to the superconducting fluctuations and thus the value of ~ 100 K deduced from the peak value, $S/T \sim 2.2 \mu\text{V}/\text{K}^2$, should be taken as the upper limit of the Fermi-temperature estimate. In addition, $T_c/T_F \sim 0.1$ may be in the crossover region of pairing condensate of fermions (Nakagawa, et al., Science 372, 190 (2021)). In the revised MS, we have added the following explanation:

[In 6th paragraph] " S/T appears to saturate on approaching T_c even below 0.5 GPa. Our previous Nernst-effect measurements suggest the enhanced preformation of the Cooper pairs particularly below 0.5 GPa¹⁹. As the saturation of S/T from the logarithmic T -dependence may be due to the superconducting fluctuations, the low-temperature limit of $-S/T$ in the normal state should be larger than the peak value of $\sim 2.2 \mu\text{V}/\text{K}^2$ at 8 K shown in Fig. 3a; namely, the Fermi temperature is lower than ~ 100 K."

Comment #2-3

3. It would be a good idea to compare the peak S/T obtained here, with other cases of quantum criticality: CeCoIn5 [$S/T > 6 \mu\text{V}/\text{K}^2$, PRL PRL 99, 147005 (2007)], YbRh2Si2 [$S/T > 8 \mu\text{V}/\text{K}^2$, PRL PRL 109, 156405 (2012)], and CeCu5.9Au0.1 [$S/T > 30 \mu\text{V}/\text{K}^2$, Physica B 259-261, 380 (1999) and Physica B 281–282, 359 (2000)]. Note that the Fermi temperature in these heavy-fermion quantum critical systems, becomes as small as a few Kelvins.

Reply #2-3

We agree that comparing the absolute values of S/T makes sense. However, there are several cases in which superconductivity masks the enhancement of S/T at low temperatures. Considering such a case and paying particular attention to the quantum criticality, we think that the coefficient, $|\gamma'|$, of the logarithmic T -dependence better characterize the material dependence of the quantum criticality. Indeed, the $|\gamma'|$ values of heavy electron systems, cuprates and the present system appear correlated with the density of states or inverse of the Fermi energy. To describe this, we added the following underlined sentences:

[In 9th paragraph] "As S/T probes the effective mass or density of states through the relation, $S \sim C/ne$, the coefficient $|\gamma'|$ of the logarithmic T -dependence is a possible measure of the strength of the quantum criticality, which appear in the specific heat or effective mass³⁸. The reported values of $|\gamma'|$ are 0.01–0.05 $\mu\text{V}/\text{K}^2$ for electron-doped cuprates²⁷, 0.05–0.16 $\mu\text{V}/\text{K}^2$ for hole-doped cuprates^{24–26}, 0.3–0.9 $\mu\text{V}/\text{K}^2$ for ion pnictides (Ba(Fe_{1-x}Co_x)₂As₂) (ref.²⁹), and 2.3, 4.5, 6.2 $\mu\text{V}/\text{K}^2$ for heavy electron systems (UCoGe, YbRh₂Si₂, and CeCu_{5.9}Au_{0.1}, respectively)^{31–33} (see Table S1 in Supplementary Information for $|\gamma'|$ values of other materials). Thus, the present $|\gamma'|$ value for κ -HgBr, $\sim 1.2 \mu\text{V}/\text{K}^2$, is situated in between the values of the cuprates and heavy electrons, reflecting the density of states or inverse of the Fermi energy."

"

Comment #2-4

4. Fig. contains an inset showing the pressure dependence of A , the prefactor of T -square

resistivity. Where is the resistivity data to see how A has been extracted and what is the temperature window of T-square resistivity?

Reply #2-4

In reply to the question above, we newly show the resistivity data used for extracting the A values in Supplementary Information (Figs. S5a and S5b). As shown there, the temperature range used depends on pressure; below ~13 K at 0.52 GPa and below ~17 K at 1 GPa. In the revised MS, we added the following underlined description:

[In 8th paragraph] “The pressure dependence of A measured with a separate sample is displayed in the inset of Fig. 3a, which exhibits its remarkable increase well before entering the non-Fermi liquid regime (*see Supplementary Information for the resistivity data*).”

Comment #2-5

5. If the system is not a Fermi liquid below 1 GPa, how come that its resistivity follows a T-square temperature dependence?

Reply #2-5

Related to this experimental feature, we wrote in the original MS a short discussion in terms of the behavior of the coefficient A as follows: “It is noted that the enhanced S/T values are not sharply suppressed upon the crossover from the non-Fermi liquid to the Fermi liquid at around 0.5 GPa. We consider this as a possible manifestation of the strong electron correlation in the marginal Fermi liquid nearby a non-Fermi liquid. The coefficient A in the temperature dependence of resistivity, $\rho = \rho_0 + AT^2$, in the Fermi-liquid regime is a measure of correlation strength or quasi-particle damping rate. The pressure dependence of A measured with a separate sample is displayed in the inset of Fig. 3a, which exhibits its remarkable increase well before entering the non-Fermi liquid regime.”

However, the discrepancy between the behaviors of S/T and resistivity in 0.5-1.0 GPa has no clear explanation. Having the reviewer’s comment, we further considered this problem and here raise conceivable origins. First, as seen in Fig. 2a in the revised MS (Figs. 2a and 2b in the previous MS are combined into the new Fig. 2a, following Reviewer#1’s suggestion), -S/T under 0.7 and 0.9 GPa shows downward deviation from the logarithmic dependence at low temperatures, suggesting possible Fermi liquid behavior well below 10 K (in the absence of superconductivity) in this pressure range. From the Boltzmann equation, the Seebeck coefficient is deduced as $S(T) \propto \int \sigma(\varepsilon)(\varepsilon - \varepsilon_F)\{-f'(\varepsilon)\}d\varepsilon$ while the conductivity is expressed as $\sigma(T) \propto \int \sigma(\varepsilon)\{-f'(\varepsilon)\}d\varepsilon$, where $\sigma(\varepsilon)$ is the energy-dependent conductivity. We note that, while the factor of $-f'(\varepsilon)$ in $\sigma(T)$ gives thermal average at $\varepsilon = \varepsilon_F$, the factor of $(\varepsilon - \varepsilon_F)\{-f'(\varepsilon)\}$ in $S(T)$ roughly gives thermal average at an energy higher (by roughly $k_B T$) than ε_F . As seen in Fig. S5a, the resistivity deviates from the quadratic temperature dependence at low temperatures towards lower powers of temperatures around 15-20 K depending on pressure. This suggests that a possible origin of the discrepancy is the difference in the energy range probed by the Seebeck coefficient and resistivity. Additionally, we do not rule out the sample dependence and possible uncertainty in pressure determination. To describe these considerations, we added the following sentences in the revised MS:

[In 8th paragraph] “Furthermore, we note that the energy state probed by S/T is higher by $\sim k_B T$ than the Fermi energy, at which the resistivity is determined. As mentioned above, the behavior of -S/T under 0.7 and 0.9 GPa implies possible Fermi liquid behavior well below 10 K. In the pressure range of 0.5-1.0 GPa, the resistivity that exhibits the T^2 dependence at low temperatures starts to deviate towards lower powers of temperature around 15-20 K (see Fig. S5a in Supplementary Information). Thus, the seeming discrepancy between the logarithmic S/T and the quadratic resistivity under pressures of 0.5-1.0 GPa may be attributable to the difference in the energy ranges relevant to the two

quantities. Sample dependence and uncertainty in pressure determination in different runs may be additional origins of the discrepancy.”

Comment #2-6

6. Is the absolute value of A reliable (despite cracks appearing in such fragile samples under pressure)? If yes, the magnitude of A can also be a guide to the Fermi temperature [See for example Fig4b in Nature Commun. 11: 3846 (2020)].

Reply #2-6

It is known that the resistivity of organic conductors has large ambiguity in the absolute value although its relative variation, e.g., against temperature or pressure is reliable. As seen in Fig. 2a, the values of the Seebeck coefficient are reproduced with different samples and thus reliable even in the absolute value. If we deduce the Fermi temperature of the present system at 1 GPa from Fig.4b in Nature Commun. 11: 3846 (2020) and the A value, the Fermi temperature turns out to be an order of magnitude smaller than the value deduced from S/T , as shown below. There is no reason that the former is more reliable than the latter; so, we refrain from discussing the Fermi temperature in terms of A.

(Figure is taken from *Nature Commun.* 11: 3846 (2020))

[Reply to Reviewer #3]

Comment #3-0

The authors performed thermoelectric measurements on the material κ -HgBr and demonstrated that the Seebeck effect exhibits features of quantum criticality. To this end, the Seebeck coefficient is studied as both a function of temperature and pressure. Interestingly, the quantum critical behavior persists for a finite region of the phase diagram. Hence, the authors propose that the criticality does not stem from a single critical point but rather an extended critical phase. Since transport in this material is essentially taking place in two dimensions, their finding would constitute the discovery of a critical phase in two dimensions, which is a rather exciting prospect. Critical phases are well known in one spatial dimension, but agreement on whether they could exist in two dimensions has currently not been achieved. The results are presented in a clear way and the paper is written in a very accessible style. I, therefore, think that the results are highly topical and deserve publication in some form. However, I would like the authors to take into account some of the following questions and remarks before I can make a final decision.

Reply #3-0

We are grateful to Reviewer #3 for reading our manuscript thoroughly and giving us enlightening comments for improving the MS. Now, taking all the comments into account, we revised the previous MS. Below, we reply point by point to all the comments and describe the revised parts, which are underlined in the following replies and colored in the MS.

Comment #3-1

1) The authors claim in the abstract that this material is currently the only candidate of a doped spin liquid known. I think this is a too bold statement to make at the present moment. There are several novel experimental platforms of doped kagome antiferromagnets and Kitaev systems that are under investigation and where quantum spin liquids are seriously considered. I suggest removing the statement that this material is the only candidate for a doped spin liquid.

Reply #3-1

Following the suggestion, we replaced the phrase, “Promising and currently the only candidate of the doped spin liquid” in the abstract (first paragraph) with “A promising candidate of the doped spin liquid”.

Comment #3-2

2) The authors should discuss theoretical proposals for critical phases in more detail. The most prominent critical spin liquid state which constitutes a phase in two dimensions is the Dirac spin liquid. While there are several theoretical proposals that this state can be realized in triangular geometries, its existence is not certain. In fact, it is as of today not known whether such a state can actually be a stable phase, e.g. <https://journals.aps.org/prl/abstract/10.1103/PhysRevLett.123.207203>.

Reply #3-2

The nature of the possible underlying spin liquid is of profound interest and the present quantum critical behavior may give a clue to it. As suggested by the reviewer, the Dirac spin

liquid is a highly possible candidate. However, since the present Seebeck effect mainly comes from mobile charge carriers, not directly in the spin degrees of freedom, we think that we should be cautious about the in-depth discussion of the underlying spin-liquid nature from the present observation alone. Moreover, the present system situated near the Mott transition has an intermediate strength of interaction; discussion in terms of the Heisenberg model as suggested above might not be plausible. Nevertheless, stimulated by the reviewer's comment, we examined other experimental results to seek for possible signatures suggestive of the spin liquid nature and came aware that the μ SR study of the present system and the nature of the superconducting phase in a related system are informative. In the next Reply#3-3, we jointly reply to Comments #3-2 and #3-3 with discussion on this line .

Comment #3-3

3) Recent theoretical proposals have studied the effect of hole doping a spin liquid in the triangular lattice Hubbard model and its thermodynamics, e.g. <https://journals.aps.org/prl/abstract/10.1103/PhysRevLett.125.157002> and <https://journals.aps.org/prx/abstract/10.1103/PhysRevX.11.041013>. The manuscript would benefit from discussing, whether the author's findings could be explained by such theoretical models.

Reply #3-3

We appreciate constructive suggestions. As the second reference (PhysRevX.11.041013) deals with the non-doped half-filled Hubbard model, we cite it as a study suggesting the complexity of competing phases on a triangular lattice and extend our discussion based on the first reference (PhysRevLett.125.157002) below and in the revised MS. The nature of the spin liquid is closely related to that of superconductivity emerging from it. The suggested case (PhysRevLett.125.157002) is a correspondence between chiral spin liquid and chiral superconductivity through doping. For the present system, a μ SR study suggested the absence of time-reversal symmetry breaking in the superconducting state, naturally leading to a conjecture that a chiral spin liquid may not underlie in the normal state. Likewise and more evidently, the superconductivity and its Gutzwiller-projected spin liquid share the symmetry. The pressure-induced Mott transition, which allows/prohibits double occupancies, can be the experimental realisation of the Gutzwiller projection. The superconductivity that appears by pressurizing the spin-liquid candidate κ -(ET)₂Cu₂(CN)₃, which is the mother system of κ -HgBr, is shown to be nodal by NMR experiments (Ref. 48). This may suggest that a Dirac spin liquid is underlying in κ -HgBr. In the revised MS, we added a new paragraph describing this discussion as follows:

[9th paragraph] *“The nature of the possible underlying spin liquid in κ -HgBr is of profound interest and the present observation potentially gives a clue to it. However, since the present Seebeck effect mainly comes from mobile charge carriers, not directly from spins, it is not straightforward to discuss the underlying spin-liquid nature from the present observation. Moreover, the Hubbard model with the intermediate strength of interaction, which is appropriate to κ -HgBr, suggests the complexity of competing phases on a triangular lattice⁴⁵. Nevertheless, superconductivity that emerges from a spin liquid is informative to the nature of the underlying spin liquid; for example, a correspondence between a chiral spin liquid and a chiral superconductivity that emerges by doping is theoretically suggested⁴⁶. A μ SR study of κ -HgBr showed the absence of time-reversal symmetry breaking in the superconducting state⁴⁷, implying less likeliness of a chiral spin liquid. Likewise, the superconductivity and its Gutzwiller-projected spin liquid share the symmetry. The pressure-induced Mott transition, which prohibits double occupancies, can be the experimental realization of the Gutzwiller projection. The superconductivity that appears by pressurizing the spin-liquid candidate κ -(ET)₂Cu₂(CN)₃, the mother system of κ -HgBr, is shown to be nodal by NMR experiments⁴⁸. A Dirac spin liquid⁴⁹ appears more*

likely.”

Ref. 45: Wietek, A. et al., *Phys. Rev. X.* **11**, 041013 (2021).

Ref. 46: Jiang, Y-F. & Jiang, H-C., *Phys. Rev. Lett.* **125**, 157002 (2020).

Ref. 47: Satoh, K. et al., *Physica B* **404**, 597 (2009).

Ref. 48: Shimizu, Y. et al, *Phys. Rev. B* **81**, 224508 (2010).

Ref. 49: Hu, S., Zhu, W., Eggert, S. & He, Y.-C. *Phys. Rev. Lett.* **123**, 207203 (2019).

REVIEWER COMMENTS

Reviewer #1 (Remarks to the Author):

Dear Editors,

I have read the authors' letter and the revised manuscript. In my opinion, it still does not meet the standards of Nature Communications. Here is a list of my comments on the authors' response:

#1-1

In my first comment, I asked whether S/T observed in κ -HgBr for a range of pressures could not be related to the presence of underlying quantum critical (QC) point rather than a QC phase. I gave the iron based superconductor $\text{Ba}(\text{Fe}_{1-x}\text{Co}_x)_2\text{As}_2$ as an example of such behaviour. The authors responded that, unlike $\text{Ba}(\text{Fe}_{1-x}\text{Co}_x)_2\text{As}_2$, the quantum spin liquid state in κ -HgBr allows for formation of the QC phase. This may be true, but the point is that even if something is in principle allowed, it does not necessarily mean that it actually occurs. I believe more solid arguments have to be presented. The statement added to the text (i.e. "The QSL that possibly extends in the low-pressure range is considered as a key to the quantum criticality persisting in a finite parameter range, not at a point.") also needs to be supported, at least with some references. The next sentence ("The extension of the present quantum criticality in a finite pressure.") sounds incomplete.

#1-2

While I agree that electrical and thermoelectric responses probe energetically different electrons, it is difficult to believe without further justification that a distance of a few meV can cause one to behave like Fermi liquid and the other like non-Fermi liquid system. Perhaps the origin of the difference might be related to a different sensitivity of ρ and S to small-angle inelastic scattering, but this needs careful consideration.

#1-3

In the original comment I asked what caused the authors to conclude that "the present results lend support to the picture of pressure-induced BEC-BCS crossover in κ -HgBr from thermoelectric point of view.". Is this about the dependence of the width of the superconducting transition on magnetic field and pressure? This may be consistent with the BEC-BCS scenario, but it can hardly be called a thermoelectric property.

I also have some doubts about the newly introduced text, namely:

#1-1R

On page 3 the authors claim that they see appreciable deviations from the logarithmic T-dependence of S/T data for $p = 0.9$ and 0.7 GPa. It is not clear to me what they mean by this, since $S/T(T)$ at $p = 0.7$ GPa (multiplied by a factor of 1.16) is virtually the same as one at $p = 0.4$ GPa (see attached figure). If the statement regards the low temperature part, the likely source of the difference is pressure-dependent T_c . By the way, the legend in Fig. 4c says that $p = 0.5$ GPa, whereas it was rather 0.65 GPa.

#1-2R

It is not clear to me what the authors are trying to say in the paragraph starting with “The nature of the possible underlying spin liquid [...]”. Why do they think that κ -HgBr hosts a Dirac spin liquid?

Reviewer #2 (Remarks to the Author):

I recommend publication.

Reviewer #3 (Remarks to the Author):

As detailed in my previous review, I find the authors finding highly topical and timely. After addressing my previous questions and suggestions in a more than appropriate way, I would now like to recommend the manuscript for publication in Nature Communications.

Reply to Reviewer #1's comments

Let us express our sincere thanks to Reviewer #1 for further reviewing our revised manuscript and pointing out still elusive descriptions. The reviewer's suggestions greatly helped us improve the manuscript. Below we reply to all comments and describe the revisions of the manuscript, which are highlighted by blue characters in the manuscript. (The red characters indicate the previously revised parts.)

Comment #1-1

In my first comment, I asked whether S/T observed in κ -HgBr for a range of pressures could not be related to the presence of underlying quantum critical (QC) point rather than a QC phase. I gave the iron based superconductor $\text{Ba}(\text{Fe}_{1-x}\text{Co}_x)_2\text{As}_2$ as an example of such behaviour. The authors responded that, unlike $\text{Ba}(\text{Fe}_{1-x}\text{Co}_x)_2\text{As}_2$, the quantum spin liquid state in κ -HgBr allows for formation of the QC phase. This may be true, but the point is that even if something is in principle allowed, it does not necessarily mean that it actually occurs. I believe more solid arguments have to be presented. The statement added to the text (i.e. "The QSL that possibly extends in the low-pressure range is considered as a key to the quantum criticality persisting in a finite parameter range, not at a point.") also needs to be supported, at least with some references. The next sentence ("The extension of the present quantum criticality in a finite pressure.") sounds incomplete.

Reply #1-1

The direct experimental verification of the quantum critical "point" is the demonstration of the fan-shaped spread of quantum criticality from a single point. In case the low-temperature part is hidden by the superconducting phase, the critical "point" can be suggested, for example, by the following experimental features:

- i) The quantum critical behavior extending in a range of control parameter (doping, pressure, etc) above T_c shows some features suggesting the existence of a critical point.
- ii) There is a symmetry-breaking second-order (or weakly first-order) transition whose temperature is pushed down toward absolute zero when the control parameter is varied.

In the case of $\text{Ba}(\text{Fe}_{1-x}\text{Co}_x)_2\text{As}_2$, the magnitude of S/T and the coefficient of its logarithmic temperature dependence exhibit clear cusp-like anomalies at a doping level of $x=0.05$ (Figs. 3 and 2, respectively, in Phys. Rev. B 87, 224508 (2013)), which is suggestive of a critical point regarding i). Regarding ii), an SDW transition line heads toward absolute zero (Fig. 4 in that reference). These observations are considered to provide good reasons to suggest the existence of a quantum critical point. In the present case of κ -HgBr, however, there are no indications of i) as observed in $\text{Ba}(\text{Fe}_{1-x}\text{Co}_x)_2\text{As}_2$ (please see Figs. 2a, 2b, and 3a in the manuscript). Regarding ii), κ -HgBr shows no phase transition even when the pressure is varied. Thus, the present experimental data have no indications supporting the critical point. Furthermore, a frustration-induced non-ordered phase as in the present system is theoretically suggested to host a quantum critical phase extended in a finite range of control parameter, not at a point (Refs. 8 and 9). For the present system, there are neither experimental features nor theoretical suggestions that support a quantum

critical point. We describe the above explanation in the revised MS as follows:

[7th paragraph] “One should be cautious about a possible case that the fanning out toward a finite temperature of quantum criticality originating from a single point is hidden by the superconducting phase. Such a case is argued, e.g., for $Ba(Fe_{1-x}Co_x)_2As_2$, in which the magnitude of S/T and the coefficient of its logarithmic temperature dependence above T_c exhibit cusp-like anomalies at a doping level and an SDW transition appears to vanish at that doping level, suggesting a symmetry-breaking quantum phase transition²⁹. These features provide good reasons to suggest the existence of a quantum critical point in $Ba(Fe_{1-x}Co_x)_2As_2$. κ -HgBr, however, shows no appreciable anomalies in the magnitude and temperature dependence in S/T in the quantum critical pressure region (Figs. 2a, 2b, and 3a), and no symmetry-breaking phase transition in the pressure range studied. Thus, the present experimental data have no indications supporting a critical point. Furthermore, frustration-induced non-ordered phases as in the present system are theoretically suggested to likely host quantum critical phases extended in a finite range of control parameter^{8,9}. Considering that other materials with spin frustration are argued to host such a phase⁴⁻⁷, spin frustration would be a key to the stabilisation of the quantum critical phase, as suggested theoretically^{8,9}.”

Following the suggestion, we have cited theoretical works (Refs. 8 and 9) which support the statement, “The QSL that possibly extends in the low-pressure range is considered a key to the quantum criticality persisting in a finite parameter range, not at a point” And, the next sentence (“The extension of the present quantum criticality in a finite pressure.”) was what we failed to delete during revision. So, we deleted it.

Comment #1-2

While I agree that electrical and thermoelectric responses probe energetically different electrons, it is difficult to believe without further justification that a distance of a few meV can cause one to behave like Fermi liquid and the other like non-Fermi liquid system. Perhaps the origin of the difference might be related to a different sensitivity of ρ and S to small-angle inelastic scattering, but this is needs careful consideration.

Reply #1-2

We agree that different scattering profiles in the force-driven and temperature-driven electron diffusions are likely responsible for the not-straightforward relationship between the resistivity and the Seebeck coefficient in the transition region from non-Fermi liquid to Fermi liquid. We would like to emphasize that the different energy scales of the two quantities can be a major origin causing the different scattering profiles. While $\rho(T)$ is determined by the thermal excitations around ε_F , $S(T)$ is expressed by $S(T) \propto (\rho(T)/T) \int \sigma(\varepsilon, T)(\varepsilon - \varepsilon_F)\{-f'(\varepsilon, T)\}d\varepsilon$, in which $(\varepsilon - \varepsilon_F)\{-f'(\varepsilon)\}$ takes a maximum at $\varepsilon = \varepsilon_F + 1.5k_B T$, meaning that the relevant energy range for S not only broadens but also **progressively shifts to higher energies as the temperature is raised**. The energy shift by $1.5 k_B T$ should cause significant effects beyond an effective temperature rise in correspondence with the behavior of electrical resistivity shown below (Fig. 2(c) in Ref. 19). To explain this, we revised the 8th paragraph as follows:

[8th paragraph] “However, the seeming discrepancy between the non-saturated S/T and the quadratic resistivity under pressures of 0.5-1.0 GPa requires further consideration,

which invokes different scattering profiles in the temperature-driven and force-driven electron diffusions. The Seebeck coefficient is expressed by $S(T) \propto (\rho(T)/T) \int \sigma(\varepsilon, T)(\varepsilon - \varepsilon_F)\{-f'(\varepsilon, T)\}d\varepsilon$ with the conductivity, $\sigma(\varepsilon, T)$, and the energy derivative of the Fermi-Dirac function, $f'(\varepsilon, T)$. We note that $(\varepsilon - \varepsilon_F)\{-f'(\varepsilon)\}$ takes a maximum away from the Fermi energy ε_F by $\sim 1.5k_B T$, meaning that the relevant energy range for S not only thermally broadens but shifts to higher energies as the temperature is raised. This progressive energy shift with the increase of temperature should cause significant effects beyond an effective temperature rise in correspondence with the behavior of electrical resistivity which shows the T^2 dependence at low temperatures and deviates towards lower powers of temperature above 15-20 K (see Fig. S5a in Supplementary Information and Ref. 19).”

Comment #1-3

In the original comment I asked what caused the authors to conclude that “the present results lend support to the picture of pressure-induced BEC-BCS crossover in κ -HgBr from thermoelectric point of view.” Is this about the dependence of the width of the superconducting transition on magnetic field and pressure? This may be consistent with the BEC-BCS scenario, but it can hardly be called a thermoelectric property.

Reply #1-3

Yes, with this sentence, we meant the pressure and magnetic-field dependences of the superconducting transition profile, as the reviewer pointed out. But, it is a repetition of previous sentences, and the wording of “thermoelectric property” is inappropriate as the reviewer suggested. So, we deleted this sentence.

Comment #1-1R

I also have some doubts about the newly introduced text, namely:

On page 3 the authors claim that they see appreciable deviations from the logarithmic T-dependence of S/T data for $p = 0.9$ and 0.7 GPa. It is not clear to me what they mean by this, since $S/T(T)$ at $p = 0.7$ GPa (multiplied by a factor of 1.16) is virtually the same as one at $p = 0.4$ GPa (see attached figure). If the statement regards the low temperature part, the likely source of the difference is pressure-dependent T_c . By the way, the legend in Fig. 4c says that $p = 0.5$ GPa, whereas it was rather 0.65 GPa.

Reply #1-1R

Our discussion in this paragraph is directed to the low-temperature range where $-S/T$ varies logarithmically with temperature at pressures below 0.65 GPa and deviates from that behavior at 0.7 GPa and 0.9 GPa as indicated by an arrow in the figure below. The downward deviations of $-S/T$ at 0.7 GPa and 0.9 GPa are clear and are the normal-state property not due to the pressure variation of T_c because T_c at 0.7 GPa and 0.9 GPa, is equal or lower than T_c at pressures below 0.65 GPa and thus cannot push $-S/T$ downward.

We appreciate the suggestion of our mislabeling in Fig.4c. We corrected it.

Comment #1-2R

It is not clear to me what the authors are trying to say in the paragraph starting with “The nature of the possible underlying spin liquid [...]”. Why do they think that κ -HgBr hosts a Dirac spin liquid?

Reply #1-2R

We added this paragraph in response to Reviewer #3's suggestion that we should discuss the nature of the underlying quantum spin liquid. Then, we received his/her evaluation, "After addressing my previous questions and suggestions in a more than appropriate way, I would now like to recommend the manuscript for publication in Nature Communications." However, having the above comment, we scrutinized this paragraph and have become aware that the relationship between the quantum spin liquid and superconductivity emerging under pressure should be described in a more intelligible way. So, we have rewritten that part as follows:

[11th paragraph] "Nevertheless, it is suggestive that a spin liquid is theoretically generated from superconductivity by Gutzwiller projection, namely, removing double occupancies and the symmetries of the two phases are mutually correspondent like d-wave superconductivity vs. Dirac (nodal) spin liquid or chiral superconductivity vs. chiral spin liquid⁴⁶. Therefore, the symmetry of superconductivity emerging from a spin liquid under pressure is quite informative because the pressure-induced Mott transition, which prohibits double occupancies, can be the experimental realization of the Gutzwiller projection. The superconductivity that appears by pressurizing the spin-liquid candidate $\kappa\text{-(ET)}_2\text{Cu}_2(\text{CN})_3$, the undoped analog of $\kappa\text{-HgBr}$, is shown to be nodal by NMR experiments⁴⁸. A Dirac spin liquid⁴⁹ appears likely. Furthermore, a μSR study of $\kappa\text{-HgBr}$ showed the absence of time-reversal symmetry breaking in the superconducting state⁴⁷, implying less likeliness of a chiral spin liquid. Anyway, the nature of the underlying spin liquid is an issue of future investigation."

REVIEWERS' COMMENTS

Reviewer #1 (Remarks to the Author):

I am satisfied with the authors' response and I think that the revised version of the manuscript is suitable for publication.